# Impact of the UK soft drinks industry levy on health and health inequalities in children and adolescents in England: An interrupted time series analysis and population health modelling study

**Linda J. Cobiac**[1], **Nina T. Rogers**[2], **Jean Adams**[2], **Steven Cummins**[3], **Richard Smith**[4], **Oliver Mytton**[5], **Martin White**[2], **Peter Scarborough**[6]*

1 School of Medicine and Dentistry, Griffith University, Queensland, Australia, 2 MRC Epidemiology Unit, University of Cambridge School of Clinical Medicine, Institute of Metabolic Science, Cambridge, United Kingdom, 3 Population Health Innovation Lab, Department of Public Health, Environments & Society, London School of Hygiene & Tropical Medicine, London, United Kingdom, 4 Faculty of Health and Life Sciences, University of Exeter, Exeter, United Kingdom, 5 UCL Great Ormond Street Institute of Child Health, University College London, London, United Kingdom, 6 Nuffield Department of Primary Care Health Sciences, University of Oxford, UK & NIHR Oxford Health Biomedical Research Centre at Oxford, Oxford, United Kingdom

* peter.scarborough@phc.ox.ac.uk

**Data Availability Statement:** The data that support the PRIMEtime model are freely available from the following sources: o Global Burden of Disease Project (https://vizhub.healthdata.org/gbd-results/) o Office for National Statistics (https://vizhub.

## Abstract

### Background

The soft drinks industry levy (SDIL) in the United Kingdom has led to a significant reduction in household purchasing of sugar in drinks. In this study, we examined the potential medium- and long-term implications for health and health inequalities among children and adolescents in England.

### Methods and findings

We conducted a controlled interrupted time series analysis to measure the effects of the SDIL on the amount of sugar per household per week from soft drinks purchased, 19 months post implementation and by index of multiple deprivation (IMD) quintile in England. We modelled the effect of observed sugar reduction on body mass index (BMI), dental caries, and quality-adjusted life years (QALYs) in children and adolescents (0 to 17 years) by IMD quintile over the first 10 years following announcement (March 2016) and implementation (April 2018) of the SDIL. Using a lifetable model, we simulated the potential long-term impact of these changes on life expectancy for the current birth cohort and, using regression models with results from the IMD-specific lifetable models, we calculated the impact of the SDIL on the slope index of inequality (SII) in life expectancy. The SDIL was found to have reduced sugar from purchased drinks in England by 15 g/household/week (95% confidence interval: −10.3 to −19.7). The model predicts these reductions in sugar will lead to 3,600 (95% uncertainty interval: 946 to 6,330) fewer dental caries and 64,100 (54,400 to 73,400)

healthdata.org/gbd-results/) o Health Survey for England (https://digital.nhs.uk/data-and-information/publications/statistical/health-survey-for-england) o Children's Dental Health Survey 2013 (https://digital.nhs.uk/data-and-information/publications/statistical/children-s-dental-health-survey/child-dental-health-survey-2013-england-wales-and-northern-ireland) o Adult's Dental Health Survey 2009 (https://digital.nhs.uk/data-and-information/publications/statistical/adult-dental-health-survey/adult-dental-health-survey-2009-summary-report-and-thematic-series) The interrupted time series analyses uses commercial data purchased from Kantar Fast Moving Consumer Goods Panel and cannot be shared without permission from the data owners. Data requests can be submitted to Kantar using the online contact us form found here: https://www.kantar.com/contact/hq-general-contact.

**Funding:** This research is supported by a project grant from the NIHR Public Health Research Programme (NIHR PHR 16/130/01). PS is supported by the NIHR Oxford Health Biomedical Research Centre at Oxford (NIHR203316). OM is supported by a UKRI Future Leaders Fellowship (MR/T041226/1). MW and JMA are supported by an intramural programme grant within the MRC Epidemiology Unit (MC/UU/00006/7). The funders had no role in study design, data collection and analysis, decision to publish, or preparation of the manuscript.

**Competing interests:** JA is an Academic Editor on PLOS Medicine's editorial board. OM is a member of the Faculty of Public Health and a co-investigator on the NIHR Healthy Weight Policy Research Unit.

**Abbreviations:** BMI, body mass index; IMD, index of multiple deprivation; QALY, quality-adjusted life year; SDIL, soft drinks industry levy; SII, slope index of inequality; SSB, sugar-sweetened beverage.

fewer children and adolescents classified as overweight or obese, in the first 10 years after implementation. The changes in sugar purchasing and predicted impacts on health are largest for children and adolescents in the most deprived areas (Q1: 11,000 QALYs [8,370 to 14,100] and Q2: 7,760 QALYs [5,730 to 9,970]), while children and adolescents in less deprived areas will likely experience much smaller simulated effects (Q3: −1,830 QALYs [−3,260 to −501], Q4: 652 QALYs [−336 to 1,680], Q5: 1,860 QALYs [929 to 2,890]). If the simulated effects of the SDIL are sustained over the life course, it is predicted there will be a small but significant reduction in slope index of inequality: 0.76% (95% uncertainty interval: −0.9 to −0.62) for females and 0.94% (−1.1 to −0.76) for males.

## Conclusions

We predict that the SDIL will lead to medium-term reductions in dental caries and overweight/obesity, and long-term improvements in life expectancy, with the greatest benefits projected for children and adolescents from more deprived areas. This study provides evidence that the SDIL could narrow health inequalities for children and adolescents in England.

## Author summary

### Why was this study done?

- The UK soft drink industry levy (SDIL) has led to reduction in household purchasing of sugar in drinks, but it is not known what implications this may have for addressing high rates of dental caries and obesity among children and adolescents in England, particularly those in the more deprived areas of the country that carry a higher burden of poor health.

### What did the researchers do and find?

- We performed a controlled interrupted time series analysis to measure the effects of the SDIL on the amount of sugar per household per week from soft drinks purchased, by index of multiple deprivation (IMD) quintile; then modelled the medium- to long-term impacts this may have on dental caries, obesity, life expectancy and quality of life for children and adolescents in England.

- We found a significant reduction in purchased sugar in England overall; with the largest absolute reductions in the 2 most deprived quintiles; a small, but significant, increase in purchased sugar in the middle quintile; and small significant reductions in sugar in the 2 least deprived quintiles.

- The modelling predicted that children and adolescents in the most deprived areas of England would experience the biggest gains in health, including fewer dental caries, less obesity, and improved quality of life and life expectancy.

- Children and adolescents in less deprived areas will also experience health benefits, but on a smaller scale, leading to a small but significant reduction in health inequality in England.

### What do these findings mean?

- The study indicates that the SDIL could improve health and narrow health inequalities for children and adolescents in England.

- The impact may be underestimated since our data only included drinks purchased and brought into the home. We additionally assumed that trends in drink purchasing before the SDIL announcement would have continued and that the effects of the SDIL on drink purchasing can be sustained.

## Introduction

A soft drinks industry levy (SDIL) was introduced in the United Kingdom in April 2018 to encourage soft drink manufacturers to reduce the amount of sugar in soft drinks available for purchase. It is a tiered levy of £0.18 per litre on drinks with between 5 g and 8 g of total sugars per 100 ml, and £0.24 per litre on drinks with 8 g or more of total sugars per 100 ml [1]. Drinks with less than 5 g per 100 ml of sugar attract no levy, and milk-based drinks, pure fruit juices, low-alcohol drinks (zero alcohol versions of drinks that are marketed as specific alternatives to high alcohol drinks, or drinks that have had alcohol removed from them until they have got less than 1.2% alcohol by volume) and those sold by importers or manufacturers with a volume of less than 1 million litres per year are exempt. Interrupted time series analyses of trends before and after the SDIL announcement in 2016 and implementation in 2018 found that the volume of soft drinks purchased increased by 188.8 ml (95% confidence interval: 30.7 to 346.9) per household per week, at 1 year after implementation, but that the amount of sugar in those drinks was 8.0 g (2.4 to 13.6) lower per household per week [2].

The SDIL provides incentives for drink reformulation with the aim of reducing population sugar intake and was primarily motivated by the UK government's plans to address the high prevalence of childhood obesity [3,4]. Free sugar intake is over double the recommended intake (<5% of total energy [5]) in children and adolescents [6], and sugar-sweetened beverages (SSBs) are a key contributor to the excess [7]. Consumption of SSBs is associated with increased weight gain in children and adolescents [8,9] as well as higher rates of dental caries [10]. In addition, there are strong and persistent socioeconomic gradients in both childhood obesity [11,12] and tooth decay in the UK [13,14]. Despite regular recommendations for action on health inequalities since the 1980s [15,16], there has been little change in key measures such as life expectancy [17]. A child born today in the most deprived areas of England can expect to live 9.4 years less if they are male or 7.6 years less if they are female, than had they been born into an area of least deprivation. In addition to having a shorter life they will also, on average, spend 12 more years in a poor state of health [18]. In recent years, improvements in life expectancy have stalled and even reversed in many areas of England, and the inequality has widened, a situation exacerbated by the COVID-19 pandemic [19–21].

A 2016 systematic review of the effect of SSB taxes on socioeconomic differences in health concluded that a tax on SSBs would likely lead to changes in population weight of a similar magnitude for all socioeconomic groups or slightly better outcomes for lower compared to higher socioeconomic groups [22]. Additionally, it concluded that SSB taxes would likely be financially regressive to a small degree (up to 1% or 0.6% of annual household income for low- and high-income households, respectively). Globally, as of May 2022, 53 countries had implemented some type of SSB tax [23], but there have been very few evaluations of the real-world impacts on different socioeconomic groups. Evaluation of an SSB tax in Mexico showed that it had led to larger reductions in purchases among lower socioeconomic groups [24,25]. But an evaluation of an SSB tax in Chile found larger reductions in purchases among higher socioeconomic groups [26,27].

In this study, we measure the impact of the UK SDIL on household sugar purchasing stratified by area-level deprivation. We then model the future impact of sugar reduction on health and health inequalities for children and adolescents in England. Over the medium-term (10 years), we model the impact of the SDIL on overweight/obesity, dental caries and quality-adjusted life years (QALYs), and over the long-term (lifetime), we model how sustained reductions in sugar consumption are likely to impact on the inequality in life expectancy and quality-adjusted life expectancy for the current birth cohort.

## Methods

### Analysing the effect of the SDIL on household weekly sugar purchasing

We conducted controlled interrupted time series analyses of sugar from drinks purchased for at-home consumption in England, following methods described by Rogers and colleagues [2]. For this study, we stratified the analyses by quintile of index of multiple deprivation (IMD)—an area-level measure of socioeconomic deprivation [28]. Due to differences in measurement of deprivation across Great Britain, we focused the stratified analyses on purchasing data for England only.

The analyses used household purchasing data collected between March 2014 and November 2019 by Kantar Fast Moving Consumer Goods panel, a market research company. In this data set, participating households are asked to scan the barcodes of food and drink purchases that are brought into the home. These purchasing data are uploaded weekly and linked to data on nutritional content. Household demographic information is updated annually, and participating households receive gift vouchers equivalent to £100 ($122; €112) annually. The panel data are in the form of a line list of purchases rather than a household study. Proprietary weights (called grossupweight) are applied at the level of the purchases rather than the households. Other information attached to the line list of purchases includes the week of purchase, product descriptions, the amount of sugar in grams per 100 mls, overall volume, and a unique household identifier. The mean weekly number of households in the panel in England was 17,717. To account for possible substitution effects, we included data on purchases of all drinks irrespective of whether they were liable for the levy or not. The drink categories included soft drinks, milk and milk-based drinks, no-added-sugar fruit juice, and drinks sold as powders. Data on purchases of toiletries (shampoo, hair conditioner, and liquid soap categories) were included as a control group for the analyses. We included weekly household purchasing of toiletries (liquid soap, shampoo, and hair conditioner) as a nonequivalent control group to account for background trends in household purchases. Toiletries were selected as a suitable control group because purchasing is unlikely to be influenced by seasonality or confounders, including socioeconomic position [29].

The IMD classification variable was determined by Kantar based on postcode of the household. IMD information was not provided for all households in the Kantar data set. The missing IMD information primarily affected data at the start of the collection period, i.e., before announcement of the SDIL. Given the possibility that the data were not missing at random and to minimise data loss, we imputed missing IMD values using multiple imputation by chained equations in R version 4.1.0. Variables in the imputation models included gross household income, social class, highest educational qualification, geographical region, age of main household member, and presence of children in the household.

We used the product names and product groups defined by Kantar to assign products to drinks or toiletries groups for analyses. The outcome variable for the drinks analyses was the average weekly weight of sugar in purchased drinks (g/household/week), which we estimated from the purchased products, their sugar content and volume, and the number of households at each weekly time point. We included weekly household purchasing of toiletries as a control group to account for background trends in household purchases. We applied a proprietary weight provided by Kantar so that the data reflected the size, demographic and socioeconomic characteristics of the population of Great Britain. For the stratified analyses, we further adjusted the outcome variable to reflect the data subgroup. This included adjustments of 0.865 (the proportion of the population of Great Britain in England) and 0.2 (the proportion of the England population in an IMD quintile).

We included data collected between March 2014 (24 months before announcement of the SDIL in March 2016) and November 2019 (19 months after implementation of the SDIL in April 2018). We ended follow-up at 19 months after SDIL implementation, rather than 24 months as initially intended, due to the possibility of changes in household purchasing behaviour with Britain's exit from the European Union in December 2019 and the subsequent COVID-19 pandemic and lockdowns in early 2020.

In the controlled interrupted time series analyses (S1 Text), we allowed for changes in both slope and level in the outcome variable [30]. Following methods described by Rogers and colleagues [2], the analyses were conducted using a controlled generalised least squares model approach, with an autocorrelation-moving average correlation structure, where the autoregressive order (p) and moving average order (q) were selected to minimise the Akaike information criterion value. In addition to binary variables for both the announcement (in March 2016 –week 108) and implementation (in April 2018 –week 214) of the SDIL, we included dummy variables to reflect changes in purchases during the months of December and January, and we included average UK monthly temperature to adjust for potential temperature-related variability in drink purchases. The impact of the SDIL on sugar purchasing was estimated from the difference between the modelled weight of sugar in purchased drinks at the end of data series (November 2019 –week 295) and the counterfactual value, which we calculated assuming there had been no announcement or implementation of the SDIL. Confidence intervals were calculated from standard errors, which we estimated using the delta method [31].

## Modelling the health impacts of sugar reduction

To model the impacts of sugar reduction on health, we constructed a lifetable model to simulate years of life lived and life expectancy of the English population (Fig 1). We ran the model with a starting population of all children and adolescents (0 to 17 years) in England in 2015 [32] and added future birth cohorts based on population projections from the Office for National Statistics [33]. At each year of simulation in the lifetable model, those alive were exposed to a risk of death (due to any cause) and experienced a quality of life, which was

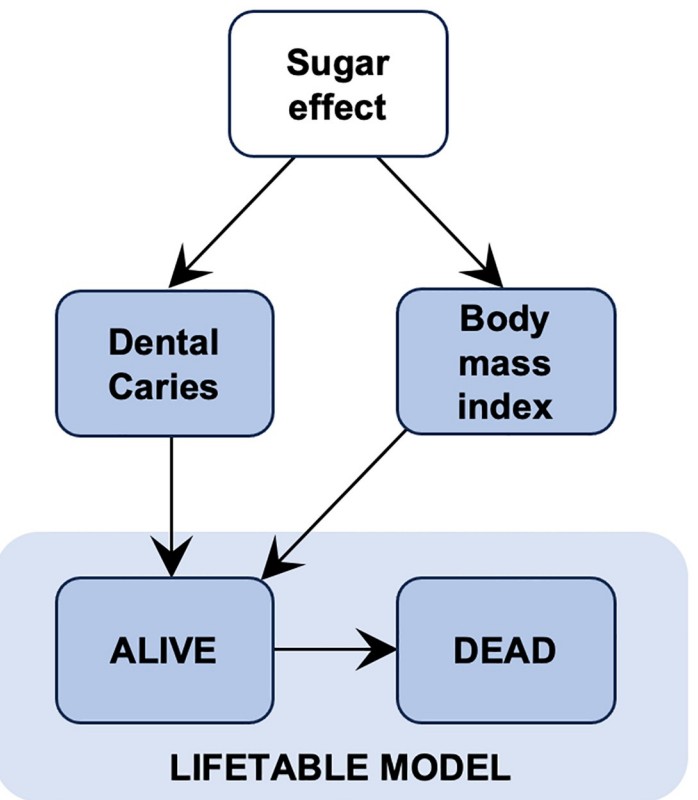

**Fig 1. The lifetable model structure.**

quantified by a utility weight ranging from 0 (death) to 1 (full health). The lifetable model was run separately by age group (0, 1–4, 5–9, 10–14. . ., 85–89, 90+), sex (male, female), and quintile of deprivation. Deprivation quintiles were based on the IMD and defined from 1 (most deprived) through to 5 (least deprived).

Fig 1 illustrates how we modelled the effects of changes in sugar purchasing on health outcomes in the lifetable model. We evaluated the population health impacts of the SDIL by simulating 2 scenarios in the lifetable model: a base case scenario, in which we assumed there had been no implementation of the SDIL, and an intervention scenario, which reflected the actual SDIL implementation in 2016 and assumed no future changes to the levy. The difference between these 2 scenarios was attributed to the sugar-reduction effects of the SDIL.

Below we provide an overview of methods used to calculate each health outcome, with additional details on the calculations and data inputs provided in an accompanying supplement (S2 Text).

From the effects on household purchasing of sugar, we estimated an average per person change in sugar consumption in each IMD quintile, assuming that food purchases are a reasonable proxy for dietary intake of sugar [34,35], and that there is an average of 2.4 people per household in England [36].

To model the impact of reduced sugar consumption on body weight, we converted sugar to calorie consumption [37] and determined the effect of reduced energy intake on body weight using energy balance equations [38,39]. Here, we assume that the reduced consumption of calories from all drinks (estimated from the interrupted time series analysis) is not associated

with changes in calorie consumption from solid foods, which is supported by evidence that calories from soft drinks have limited impact on satiety [40], on observed elasticities of demand for soft drinks [41], and on the previous interrupted time series analyses of the SDIL for Great Britain, which showed no overall substitution to confectionary products [2]. From this we determined change in body mass index (BMI) and prevalence of overweight and obesity using baseline data on the distribution of BMI by age, sex, and IMD quintile from the Health Survey for England [42]. Overweight and obesity were defined using International Obesity Taskforce definitions [43].

To model the impact of reduced sugar consumption on dental caries, we estimated the reduction in decayed, missing, and filled teeth that are deciduous (dmft) or permanent (DMFT) based on a dose-response relationship between sugar intake and dental caries [44]. Baseline dmft and DMFT rates were derived, by age and IMD quintile, from data collected in the Children's Dental Health Survey 2013 and the Adult Dental Health Survey 2009 [45,46].

The mortality rates in the lifetable model influence how many people are alive at each year of a simulation. We modelled the impact of changes in BMI on all-cause mortality rates, by age, sex, and IMD quintile, using hazard ratios derived from meta-analysis of prospective cohorts studies [47]. From these calculations we determined impact of the SDIL on life years and life expectancy by IMD quintile.

We modelled the impact of the SDIL on QALYs and quality-adjusted life expectancy using health state utility weights. We applied utility weights to reflect background quality of life by age, sex, and IMD quintile, from an analysis of EQ-5D data from the Health Survey for England [48]. Impacts on quality of life from changes in BMI and dental caries were determined using health state utilities for these conditions [49–51].

## Simulation of medium-term health impacts

We ran the lifetable model over a ten-year time horizon (2015 to 2025) to determine medium-term impacts of the SDIL on dental caries, prevalence of overweight/obesity and QALYs. For these analyses, we ran an open cohort simulation to determine outcomes for all children and adolescents who were alive or born within the ten-year time horizon. In an open cohort analysis, new birth cohorts are added as time progresses, so that at each time point the modelled population represents the whole population of children and adolescents alive at that point in time.

## Simulation of long-term health impacts

We modelled long-term impacts of the SDIL on life expectancy and quality-adjusted life expectancy by running a closed cohort simulation in the lifetable model. A closed cohort analysis only follows up population cohorts present at the start of the simulation (i.e., no future birth cohorts are added through time). In this simulation, we followed a cohort of all babies born in 2015 until all had died or reached 100 years of age. We assumed that the reduction in sugar consumption, which we derived from the interrupted time series analyses estimates of changes in sugar purchased in drinks, would be maintained across the life course.

To model the long-term impact of the SDIL on health inequality, we estimated the change in slope index of inequality (SII) [52], which is a commonly used metric of health inequality in England [53]. An SII reflects the difference in life expectancy between the most and least deprived groups in the population, thus an increase in SII reflects a widening of health inequalities, while a decrease in SII reflects a narrowing of health inequalities [54]. We calculated the SII by linear regression of life expectancy or quality-adjusted life expectancy (outcome variable) and IMD quintile (predictor variable). The regression coefficient reflected the SII.

## Probabilistic sensitivity analysis

We performed probabilistic sensitivity analyses to calculate 95% uncertainty intervals around all modelled estimates, drawing from uncertainty distributions around analytical input parameters (hazard ratios, health state utilities, and sugar dose-response relationships) and the SDIL effect in reducing sugar. Further details on these input parameters can be found in an accompanying supplement (S2 Text).

## Results

### Impact on purchased sugar in drinks

Table 1 shows the mean weight of sugar purchased in drinks per household per week in England in the week before SDIL announcement, in the week before SDIL implementation, and at the end of study. Before announcement of the SDIL, the mean weekly weight of purchased sugar was highest in the 2 most deprived IMD quintiles. Sugar purchasing declined in all groups over the course of the study, but the 2 most deprived IMD quintiles were still observed to have the highest levels of sugar purchasing at the end of the study.

Overall, there was a 15.0 g/household/week (95% confidence interval: 10.3 to 19.7) reduction in purchased sugar in England, compared to the counterfactual scenario in which it was assumed there had been no announcement or implementation of the SDIL (Table 1 and S3 Text). The simulated effect of the SDIL varied by quintile of deprivation. The largest absolute reductions in purchased sugar were observed in the 2 most deprived quintiles; there was a small, but significant, increase in purchased sugar in the middle quintile; and small significant reductions in sugar in the 2 least deprived quintiles.

S3 Text presents the same results, but without imputation of missing data. The household purchasing data set had a high level of missing IMD values at the beginning of the study period, but missing data were much lower by the end of the study period. Therefore, without imputation of missing data, the level of sugar purchasing appears to be increasing over time, inflating the difference between the observed and counterfactual trends, and leading to an overestimate of the effects of the SDIL on sugar purchasing. With imputation of the missing data, this inflation of the difference between counterfactual and observed data disappears, giving a more accurate (and conservative) estimate of the SDIL effect on sugar purchasing.

**Table 1. Absolute change in purchased sugar from drinks.**

| IMD | Weight of sugar in purchased drinks (g/household/week) | | | | |
|---|---|---|---|---|---|
| | Observed mean in the week prior to SDIL announcement (week 108) | Observed mean in the week prior to SDIL implementation (week 214) | Observed mean at study end (week 295) | Predicted counterfactual* mean at study end (week 295) | Difference in mean sugar at study end (week 295) |
| Q1 | 354.0 | 321.7 | 282.1 | 319.6 | −37.5 (−31.5, −43.5) |
| Q2 | 372.3 | 353.2 | 308.5 | 337.7 | −29.2 (−23.4, −35.0) |
| Q3 | 339.9 | 319.8 | 271.4 | 263.5 | 7.98 (2.23, 13.7) |
| Q4 | 332.8 | 302.1 | 267.7 | 271.4 | −3.76 (1.85, −9.37) |
| Q5 | 308.5 | 285.7 | 255.0 | 265.9 | −10.9 (−5.72, −16.0) |
| All quintiles | 341.6 | 316.4 | 276.9 | 291.9 | −15.0 (−10.3, −19.7) |

\* Counterfactual scenario is based on pre-announcement trends and therefore assumes there has been no announcement or implementation of the SDIL.

NB. IMD quintiles: Q1 (most deprived)–Q5 (least deprived). Values are mean and 95% confidence intervals.

IMD, index of multiple deprivation; SDIL, soft drinks industry levy.

**Table 2. Predicted impact on number of dental caries and cases of overweight and obesity, in children and adolescents, in the first 10 years following implementation of the SDIL.**

| IMD | Dental caries (number of dmft/DMFT) | Overweight (number of cases) | Obese (number of cases) |
|---|---|---|---|
| Q1 | −2,240 (−3,990, −579) | −22,800 (−26,400, −19,200) | −13,000 (−15,100, −11,000) |
| Q2 | −1,270 (−2,230, −325) | −18,000 (−21,500, −14,500) | −7,850 (−9,350, −6,340) |
| Q3 | 280 (37.3, 627) | 4,350 (1,220, 7,420) | 1,770 (497, 3,030) |
| Q4 | −101 (−310, 52.8) | −1,660 (−4,140, 857) | −554 (−1,380, 287) |
| Q5 | −274 (−550, −62.2) | −4,920 (−7,250, −2,610) | −1,470 (−2,160, −779) |
| All quintiles | −3,600 (−6,330, −946) | −43,000 (−49,500, −36,200) | −21,100 (−24,000, −18,200) |

NB. IMD quintiles: Q1 (most deprived)–Q5 (least deprived). Values are mean and 95% uncertainty intervals.

IMD, index of multiple deprivation; SDIL, soft drinks industry levy.

## Medium-term health impacts in children and adolescents

In the first 10 years following implementation of the SDIL, the model predicts there will be an annual average reduction of 3,600 (95% uncertainty interval: 946 to 6,330) dental caries and 64,100 (54,400 to 73,400) children and adolescents classified as overweight or obese (Table 2). This equates to a 0.59 percentage point (0.50 to 0.68) reduction in prevalence of overweight and obesity. The simulated effects are similar for males and females (S3 Text).

The total impact on quality of life is a 19,500 QALY (14,800 to 24,600) health gain for children and adolescents in England in the first 10 years after implementation of the SDIL. Due to the unequal effect of the SDIL on sugar purchasing, the health benefits are not equally distributed across all levels of deprivation. The most deprived quintiles are expected to benefit the most from improvements in health (Table 3). There are large significant health gains in the 2 most deprived quintiles (Q1: 11,000 QALYs [8,370 to 14,100] and Q2: 7,760 QALYs [5,730 to 9,970]), a small significant health loss for the middle quintile (Q3: −1,830 QALYs [−3,260 to −-501]), nonsignificant health gain for the less deprived quintile (Q4: 652 QALYs [−336 to 1,680]), and a small significant health gain for the least deprived quintile (Q5: 1,860 QALYs [929 to 2,890]).

## Long-term impacts on health and health inequalities

If the SDIL impacts on sugar purchasing are sustained over the life course the model predicts small changes in life expectancy and quality-adjusted life expectancy in England (Table 4).

**Table 3. Predicted impact on QALYs, in children and adolescents, in the first 10 years following implementation of the SDIL.**

| IMD | Change in quality-adjusted life years | | |
|---|---|---|---|
| | Female | Male | Total |
| Q1 | 5,980 (4,630, 7,550) | 5,060 (3,720, 6,560) | 11,000 (8,370, 14,100) |
| Q2 | 4,030 (3,010, 5,130) | 3,740 (2,710, 4,860) | 7,760 (5,730, 9,970) |
| Q3 | −937 (−1,660, −259) | −893 (−1,600, −242) | −1,830 (−3,260, −501) |
| Q4 | 368 (−187, 946) | 284 (−149, 741) | 652 (−336, 1,680) |
| Q5 | 1,030 (522, 1,580) | 830 (405, 1,310) | 1,860 (929, 2,890) |
| All quintiles | 10,500, (8,120, 13,000) | 9,020 (6,640, 11,500) | 19,500 (14,800, 24,600) |

NB. IMD quintiles: Q1 (most deprived)–Q5 (least deprived). Values are mean and 95% uncertainty intervals.

IMD, index of multiple deprivation; QALY, quality-adjusted life year; SDIL, soft drinks industry levy.

**Table 4. Average population increase in life expectancy and quality-adjusted life expectancy by IMD.**

| IMD | Change in life expectancy (days) | | Change in quality-adjusted life expectancy (days) | |
|---|---|---|---|---|
| | Female | Male | Female | Male |
| Q1 | 18 (15, 21) | 31 (26, 36) | 25 (20, 30) | 32 (26, 39) |
| Q2 | 16 (13, 19) | 21 (17, 25) | 21 (16, 26) | 23 (18, 29) |
| Q3 | −3.3 (−5.7, −0.94) | −5.3 (−9.1, −1.5) | −5 (−8.7, −1.4) | −6.1 (−11, −1.7) |
| Q4 | 1.3 (−0.65, 3.2) | 2.4 (−1.2, 5.9) | 2.1 (−1.1, 5.2) | 2.7 (−1.4, 6.9) |
| Q5 | 4.0 (2.1, 5.8) | 6.6 (3.5, 9.8) | 6.2 (3.2, 9.3) | 7.9 (4.1, 12) |

NB. IMD quintiles: Q1 (most deprived)–Q5 (least deprived). Values are mean and 95% uncertainty intervals. IMD, index of multiple deprivation.

Due to the unequal impacts of the SDIL on sugar purchasing across IMD quintiles, the life expectancy and quality-adjusted life expectancy impacts are not equally distributed across all levels of deprivation. The pattern is like the predicted distribution of QALY health gains and losses across the IMD quintiles; the model predicts the largest increases in life expectancy and quality-adjusted life expectancy in the 2 most deprived quintiles (Q1 and Q2), a small but significant decrease in the middle quintile (Q3), no significant effect in the less deprived quintile (Q4), and a small significant increase for the least deprived quintile (Q5).

Overall, the model predicts small but significant decreases in the SII in life expectancy and quality-adjusted life expectancy (Fig 2). For females, there is a 0.76 percentage point reduction ([95% uncertainty interval: −0.9 to −0.62]; −4.2 days [−5.0 to −3.5]) in SII for life expectancy and 0.54 percentage point reduction ([−0.67 to −0.42]; −5.7 days [−7.0 to −4.4]) in SII for quality-adjusted life expectancy. For males, there is a 0.94 percentage point reduction ([−1.1 to −0.76]; −6.6 days [−8.0 to −5.4]) in SII for life expectancy and 0.60 percentage point reduction ([−0.74 to −0.46]; −6.8 days [−8.5 to −5.3]) in SII for quality-adjusted life expectancy.

## Discussion

The UK government introduced the SDIL to address the high prevalence of childhood obesity [3,4]. Our modelling suggests that in the first 10 years after implementation, the SDIL will lead to substantial reductions in obesity and dental caries, and an overall health gain of 19,500 QALYs (0.56 QALYs per 1,000 population) for children and adolescents in England. The changes in sugar purchasing and predicted impacts on health are largest for children and adolescents in the most deprived areas (+11,000 QALYs in Q1 and +7,760 QALYs in Q2), while children and adolescents in less deprived areas will experience much smaller simulated effects (−1,830 QALYs in Q3, nonsignificant changes in Q4, and +1,860 QALYs in Q5). If the simulated effects of the SDIL are sustained over the life course, it is predicted that the cohort of children born today will experience a small (<1 percentage point) but significant reduction in health inequality.

Our finding of a 15.0 g/household/week (95% confidence interval: 10.3 to 19.7) reduction in sugar purchased in drinks in England is somewhat larger than the 8.0 g/household/week (2.4 to 13.6) reduction found by Rogers and colleagues [2] for the whole of Great Britain. The controlled interrupted time series analyses methods are identical between the 2 studies, although the data sets are slightly different. While our study of data from England included follow-up data to 19 months after SDIL implementation and the earlier study of data from Great Britain included follow-up data only to 12 months, reanalysis of the sugar reduction for Great

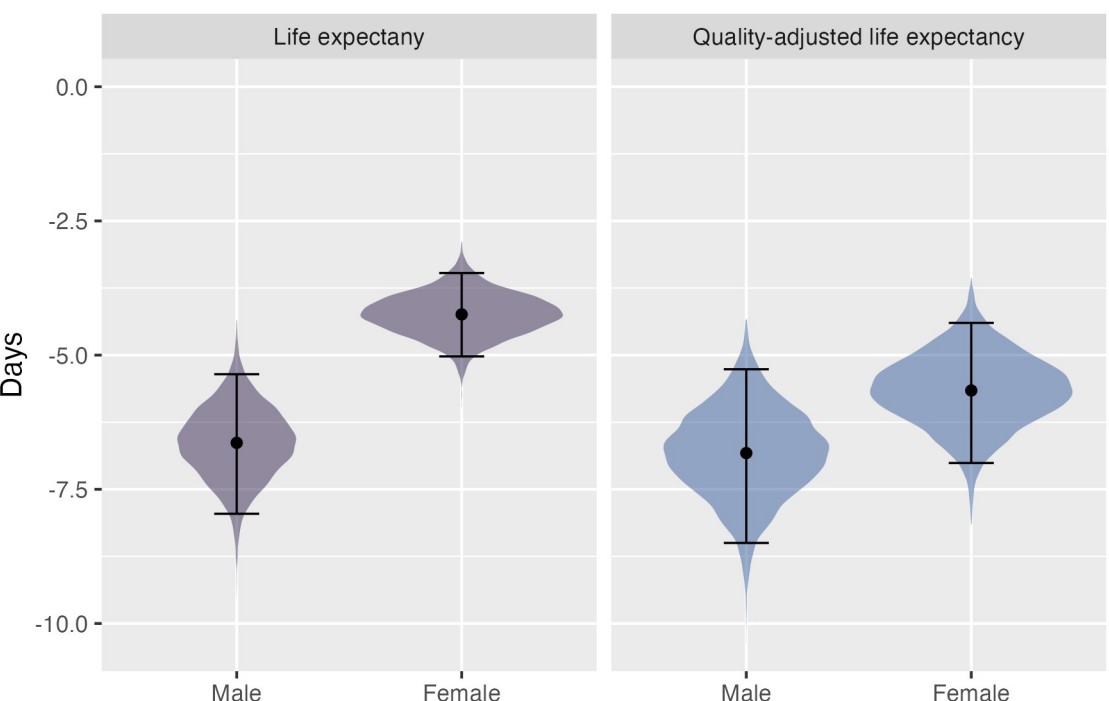

**Fig 2. Change in slope index of inequality for life expectancy (purple) and quality-adjusted life expectancy (blue).** The shading reflects the distribution of the data, the dot symbol shows the mean, and the error bars show the 95% uncertainty interval.

Britain with 19 months of follow-up data (results not yet published) indicates that the effect has remained stable between 12 and 19 months. This suggests that the difference in results are not due to differences in follow-up time since SDIL implementation. It is possible that there are regional differences in impact of the SDIL. We included data from households in England only, due to differences in the way that deprivation is measured across the different countries of Great Britain. However, England constitutes around 85% of the population of Great Britain and would be expected to have the majority influence on the results. The differences in mean sugar reduction is perhaps most likely due to the interrupted time series approach; in particular, the way in which it generates counterfactuals that act as the baseline against which we measure effect size. Small deviations in the trend in the "before" period could result in quite large deviations in the counterfactual with extrapolation over longer and longer time periods. In our analyses, the "before" period ends with the SDIL announcement in March 2016 and then the counterfactual is extrapolated over more than 3 years. A small difference in the initial downwards trend in sugar purchasing (e.g., +/− 0.1 g/month) could result in a big difference in the counterfactual estimate of sugar purchasing by the end of the study in November 2019 (+/− 4.4 g sugar). Given that the initial trend is based on regression analysis over relatively few time points, this approach is potentially going to be sensitive to small fluctuations in the initial data set (such as reducing the data set from Great Britain to England only). This may also explain the slightly unusual patterns in the sugar reduction effect across IMD quintiles, i.e., large reductions in Q1 and Q2, small significant increase in Q3, nonsignificant change in Q4 and small reduction in Q5. A similar pattern was found in analyses of the SDIL and childhood obesity [55], where a reduction in population prevalence of obesity was found in Year 6 girls in the most deprived groups (IMD 1 and 2) and the least deprived group (IMD 5), but not in the

intermediate deprivation groups (IMD 3 and 4). Other reasons for the pattern could include changing demographics of the population over the course of the study, which may have led to more noise, or perhaps differences in susceptibility to marketing or advertising. It is also important to remember that the Kantar data set includes only products brought into the home. It is possible that the proportion of in-home and out-of-home purchasing of soft drinks has been changing over time, and that this has varied by level of deprivation.

Our study suggests that the SDIL will have comparable or even larger simulated effects than other interventions in the UK for which health inequalities related to obesity and related health impacts have been evaluated. A modelling study of the NHS health checks program, a cardio-vascular risk screening program for middle-aged adults in England, estimated that it would increase life expectancy by 4.4 days for adults in the most deprived IMD quintile [56]. This is considerably less than the additional 18 days for females and 31 days for males that our model-ling predicts for the most deprived quintile, if the effects of the SDIL are sustained. It is likely that this difference is partly due to the effect size of the SDIL and partly due to its operation over the entire life course. Another modelling study that simulated removal of all unhealthy food advertising before 9 PM on UK television estimated that reductions in overweight and obesity would be 2 to 2.5 times higher for children in the lowest social grade, compared to chil-dren in the highest social grade [57]. Overall, that study estimated that the advertising restric-tions might lead to 120,000 (95% UI 34,000 to 240,000) fewer children classified as overweight or obese in the UK. The uncertainty interval around this simulated effect is large owing to the necessary extrapolation of kilocalorie effects from small-scale experimental studies. Allowing for the smaller size of the population of England (approximately 0.865 of the UK population), the simulated effects of the TV advertising restrictions are comparable to our estimated reduc-tion of 64,100 (95% uncertainty interval: 54,400 to 73,400) fewer children and adolescents clas-sified as overweight or obese after implementation of the SDIL in England.

Our study of the SDIL in England adds to the small number of evaluations of the impact of SSB taxes across different socioeconomic groups. Like the SSB tax in Mexico [24,25], our study found larger impacts for the more deprived or lower socioeconomic groups. But this differs from the experience in Chile, where the SSB tax led to larger reductions in purchases for high socioeconomic groups [26,27]. Modelling studies that have explored socioeconomic differ-ences of existing or hypothetical SSB taxes have also shown mixed results [58], while all predict beneficial health outcomes in the population overall, only 3 studies have predicted that the health effects would be progressive [59–61], as we have found, while 4 studies have predicted similar or mixed outcomes across socioeconomic groups [41,62–64].

A key strength of our study of the SDIL and health inequalities is that it combines medium-term and long-term modelling of health outcomes with an empirical evaluation of a real-world intervention. A number of previous studies have modelled socioeconomic differentials in health effects of hypothetical taxes targeting SSBs and have used economic models of demand to estimate how consumers would change purchasing behaviour for both targeted and non-targeted food and drinks [41,59–64]. In contrast, a real-world evaluation provides evidence of effects regardless of the mechanisms of change, likely increasing internal validity and provid-ing stronger evidence for policy makers [58]. For example, it is known that the SDIL had a sub-stantial impact on reformulation of soft drinks [65], which is a mechanism that would not be accounted for in studies that estimate sugar purchases using economic models of demand alone.

However, our study still relies on a number of assumptions. For example, in evaluating the impact of the SDIL on sugar in purchased drinks using the controlled interrupted time series models, we assume that modelled trends in sugar purchasing up to the time of the SDIL announcement would have continued if the SDIL had not been announced or implemented.

The data show that sugar purchased in drinks was already declining across all IMD quintiles before announcement of the SDIL in March 2016 (S2 Text), possibly due to raised awareness brought about by the focus on sugar reduction in the Childhood Obesity Plan. It is possible that these downward trends would not have continued at the same rate, particularly in the less deprived quintiles in which the mean purchase of sugar in drinks had already declined to relatively lower levels by the time of the SDIL announcement.

The Kantar data set that we used for the interrupted time series analyses is a database of products purchased and brought into the home; it does not provide information about the use or consumption of products by individuals within the household. Thus, there is uncertainty about how the changes in household purchasing of sugar in drinks are distributed among household members. In modelling the health implications from the changes in sugar attributed to the SDIL, we estimated an average per person impact on sugar intake from the average household size, within each IMD quintile, but it is likely that there will be additional variation in effects by age that we have not accounted for. Additionally, we assume that the reduction in sugar in purchased drinks translates directly into a reduction in sugar consumption. While there is some evidence to suggest that food purchases are a reasonable proxy for dietary intake of nutrients, such as sugar [34,35], there is still uncertainty around whether factors such as waste and out-of-home purchasing vary across deprivation quintiles. We also assume that there are no compensatory changes in energy expenditure (e.g., through changes in physical activity). However, we can be more confident that compensatory changes in energy intake have not occurred. Analyses of confectionery purchases over the time period of announcement and implementation of the SDIL found no evidence of compensatory changes in intake [2], which supports findings from other studies examining potential food substitution with SSB taxes [66,67]. Our purchasing data included all drink purchases that were brought into the home, but does not include drinks that were purchased and consumed outside of the home. It is estimated that 10% to 12% of expenditure on cold non-alcohol beverages in the UK is spent on out-of-home purchases [68], and it is possible this may affect some age groups more than others (e.g., teenager may be more likely to purchase drinks outside of the home). If the SDIL provoked a similar reduction in purchasing of these out-of-home drinks, then we will have underestimated the full effect of the levy—however, more work is needed to establish this.

Our modelling approach accounted for parametric uncertainty around key model parameters (drinks purchasing, sugar impact on dental caries/BMI, and dental caries/BMI impact on quality of life and mortality). However, all modelling studies are subject to structural uncertainty—that is, uncertainty about whether our modelling framework is an accurate representation of reality. Obesity is complex to model, particularly at the level of the individual. However, it is important to note that in this paper we are chiefly concerned with trends in BMI at the population level, where much of the individual-level complexity is cancelled out. This is illustrated by looking at trends in BMI distributions [69], which are smooth and predictable. This type of lifetable modelling has a long history of being used to model the long-term health effects of population changes in body weight [57,70–77]. Our long-term models necessarily make assumptions about the long-term resilience of the effect of the SDIL, which has not yet been established. The uncertainty estimates around our long-term results do not reflect this. Finally, in the quality-adjustment of life expectancy and years of life lived, we rely on published estimates of utility values to reflect quality of life associated with having dental caries or obesity, and to reflect the average background quality of life at different ages, sex, and IMD quintiles. The development and use of measurement instruments, such as the EQ-5D is well-established in the adult population, but there is far less evidence relating to the experience of children and adolescents. In their systematic review of obesity-related utility values in children and adolescents, for example, Brown and colleagues [49] highlighted the dearth of

relevant studies and the resulting high heterogeneity in meta-analysis results. While the uncertainty in child and adolescent utility values does not affect our modelled estimates of dental caries numbers, obesity prevalence, or life expectancy, our estimates of quality-adjusted life years and quality-adjusted life expectancy should be interpreted with more caution.

Our analyses show that the SDIL has had larger benefits for more deprived groups in England, and that this is likely to lead to small but significant reductions in health inequalities in the medium and long term. There are clear benefits from addressing health inequalities early in life; social gradients that develop in childhood can impact across the life course [78,79]. Recommendations to address health inequalities were made at the conclusion of the Marmot review [16] in 2010. But while there are notable examples of subsequent action at the local government level, commitment is lacking at the national level, and over 10 years on from the review, health inequalities have continued to widen [17]. Our study illustrates how the SDIL is likely to contribute to narrowing health inequalities in England.

The SDIL approach to fiscal beverage policy provides incentives for reformulation; there may be benefits from extending this approach to a wider array of products in the UK [80]. This could include sweetened milk-based drinks, which are currently excluded from the SDIL, or snack foods. While soft drinks are responsible for 10% of free sugar intake in 4 to 10 year olds and 23% in 11 to 18 year olds, sugar, preserves, and confectionery are responsible for a further 24% of intake in 4 to 10 year olds and a further 22% of intake in 11 to 18 year olds [7]. Modelling suggests that price increases in high sugar snacks might further reduce health inequalities in the UK [81]. There may be benefits in targeting non-core foods, such as confectionery, where untaxed (or subsidised) substitutes are available, to minimise any potential regressive effects of the policy. A tax on sugar was recently advocated in an independent report to inform the UK's national food strategy [82]. Modelling estimates suggest it would have a beneficial effect in reducing sugar consumption overall in the UK population [83], but further work is needed to determine if it would also reduce health inequalities.

While our study shows that the SDIL is likely to be progressive in its simulated effects on health, it does not tell us anything about the financial impact on households. But work is ongoing to examine the socioeconomic effects of the SDIL in more detail, including evaluation of household economic impacts and effects of the SDIL on the wider UK economy. Our work here focussed on the health impacts of the current cohort of children and adolescents in the UK and on the simulated effect of the SDIL on health inequalities. Further ongoing modelling work will report on the health impact on adults, including disease-specific outcomes.

## Supporting information

**S1 Text. Interrupted time series analysis.**
(DOCX)

**S2 Text. Lifetable analysis data inputs.**
(DOCX)

**S3 Text. Additional results.**
(DOCX)

## Author Contributions

**Conceptualization:** Linda J. Cobiac, Nina T. Rogers, Jean Adams, Steven Cummins, Richard Smith, Oliver Mytton, Martin White, Peter Scarborough.

**Formal analysis:** Linda J. Cobiac, Nina T. Rogers.

**Funding acquisition:** Jean Adams, Steven Cummins, Richard Smith, Oliver Mytton, Martin White, Peter Scarborough.

**Investigation:** Linda J. Cobiac.

**Methodology:** Linda J. Cobiac, Nina T. Rogers, Jean Adams, Steven Cummins, Richard Smith, Oliver Mytton, Martin White, Peter Scarborough.

**Supervision:** Peter Scarborough.

**Validation:** Linda J. Cobiac.

**Visualization:** Linda J. Cobiac, Nina T. Rogers.

**Writing – original draft:** Linda J. Cobiac.

**Writing – review & editing:** Linda J. Cobiac, Nina T. Rogers, Jean Adams, Steven Cummins, Richard Smith, Oliver Mytton, Martin White, Peter Scarborough.

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
