## [Editor Report · Decision Letter 0]

13 Oct 2023

Dear Dr Scarborough, 

Thank you for submitting your manuscript entitled "Impact of the UK Soft Drinks Industry Levy on health and health inequalities in children and adolescents in England: an interrupted time series analysis and population health modelling study" for consideration by PLOS Medicine.

Your manuscript has now been evaluated by the PLOS Medicine editorial staff and I am writing to let you know that we would like to send your submission out for external peer review.

Please re-submit your manuscript within two working days, i.e. by Oct 17 2023 11:59PM.

Sincerely,

Philippa Dodd, MBBS MRCP PhD

PLOS Medicine

---

## [Decision Letter · Decision Letter 1]

23 Nov 2023

Dear Dr. Scarborough,

Thank you very much for submitting your manuscript "Impact of the UK Soft Drinks Industry Levy on health and health inequalities in children and adolescents in England: an interrupted time series analysis and population health modelling study" (PMEDICINE-D-23-02980R1) for consideration at PLOS Medicine. 

[LINK]

In light of these reviews, I am pleased to tell you that we would like to consider a revised version that addresses the reviewers' and editors' comments. Obviously we cannot make any decision about publication until we have seen the revised manuscript and your response, and we plan to seek re-review by one or more of the reviewers. 

We expect to receive your revised manuscript by Dec 14 2023 11:59PM. Please email us (plosmedicine@plos.org) if you have any questions or concerns.

We look forward to receiving your revised manuscript. 

Best wishes,

Pippa

Philippa Dodd, MBBS MRCP PhD

PLOS Medicine

plosmedicine.org

pdodd@plos.org

COMMENTS FROM THE ACADEMIC EDITOR

Some of the queries point out fairly serious error, e.g using areal data to ascribe to each household Socio-economic status. I think this calls for a major revision. Some of the comments are not minor. If they address all the comments, we can then make a decision.

COMMENTS FROM THE EDITORS

GENERAL

Please respond to all editor and reviewer comments detailed below in full.

We understand the need to update the ITSA and to stratify by additional parameters to answer the specific question posed here. We agree with the statistical reviewer (please see below) that from a novelty perspective, it would be helpful to understand the rationale for the ITSA analysis destined for the BMJOpen as well as that presented here. Was the former to set a precedent for the (slightly updated) methodology used here, for example? Please provide details and please incorporate your response into the manuscript as necessary.

DATA AVAILABILITY STATEMENT

Thank you for including a statement pertaining to your data availability. The Data Availability Statement (DAS) requires revision. For each data source used in your study: 

We think that point c) would be most applicable here.

COMPETING INTERESTS

All authors must declare their relevant competing interests per the PLOS policy, which can be seen here:

https://journals.plos.org/plosmedicine/s/competing-interests

For authors with ties to industry, please indicate whether any of the interests has a financial stake in the results of the current study.

Please add this statement to the manuscript's Competing Interests: "JA is an Academic Editor on PLOS Medicine's editorial board."

ETHICS STATEMENT

As your study utilizes human participant data, we think an ethics statement would be appropriate. 

ABSTRACT

Thank you for formatting your abstract according to PLOS Medicine’s style. 

Abstract methods and findings:

Please include the study design, population and setting, number of participants, years during which the study took place, length of follow up, and please explicitly state the main outcome measures.

Please provide (brief) details of the databases used to leverage your data.

Please quantify the main results with 95% CIs (or UIs in this case) and p values. When reporting p values please report as p<0.001 and where higher the exact p value as p=0.002, for example. If not reporting p values, for the purpose of transparent data reporting please clearly state the reasons why not.

Please include any important dependent variables that are adjusted for in the analyses.

In the last sentence of the Abstract Methods and Findings section, please describe the main limitation(s) of the study's methodology.

Abstract Conclusions:

Please address the study implications without overreaching what can be concluded from the data; the phrase "In this study, we observed ..." may be useful.

Please interpret the study based on the results presented in the abstract, emphasizing what is new without overstating your conclusions.

Please avoid vague statements such as "these results have major implications for policy/clinical care". Mention only specific implications substantiated by the results.

Please avoid assertions of primacy ("We report for the first time....")

Line 49 – suggest ‘could’ instead of ‘will’.

AUTHOR SUMMARY

At this stage, we ask that you include a short, non-technical Author Summary of your research to make findings accessible to a wide audience that includes both scientists and non-scientists. The authors summary should consist of 2-3 succinct bullet points under each of the following headings:

• Why Was This Study Done? Authors should reflect on what was known about the topic before the research was published and why the research was needed.

• What Did the Researchers Do and Find? Authors should briefly describe the study design that was used and the study’s major findings. Do include the headline numbers from the study, such as the sample size and key findings. 

• What Do These Findings Mean? Authors should reflect on the new knowledge generated by the research and the implications for practice, research, policy, or public health. Authors should also consider how the interpretation of the study’s findings may be affected by the study limitations. In the final bullet point of ‘What Do These Findings Mean?’, please describe the main limitations of the study in non-technical language.

The Author Summary should immediately follow the Abstract in your revised manuscript. This text is subject to editorial change and should be distinct from the scientific abstract. Please see our author guidelines for more information: https://journals.plos.org/plosmedicine/s/revising-your-manuscript#loc-author-summary

METHODS and RESULTS

Of all authors who submit modelling studies we ask that the following points are included in the main manuscript. 

Please review the list below, derived from Geoffrey P Garnett, Simon Cousens, Timothy B Hallett, Richard Steketee, Neff Walker. Mathematical models in the evaluation of health programmes. (2011) Lancet DOI:10.1016/S0140-6736(10)61505-X and ensure that each item is included:

* Please provide a diagram that shows the model structure, including how the disease natural history is represented, the process and determinants of disease acquisition, and how the putative intervention could affect the system.

*Please provide a complete list of model parameters, including clear and precise descriptions of [the meaning of each parameter, together with the values or ranges for each, with justification or the primary source cited, and important caveats about the use of these values noted].

*Please provide a clear statement about how the model was fitted to the data [including goodness-of-fit measure, the numerical algorithm used, which parameter varied, constraints imposed on parameter values, and starting conditions].

*For uncertainty analyses, please state the sources of uncertainties quantified and not quantified [can include parameter, data, and model structure].

*Please provide sensitivity analyses to identify which parameter values are most important in the model. Uncertainty estimates seek to derive a range of credible results on the basis of an exploration of the range of reasonable parameter values. The choice of method should be presented and justified.

*Please discuss the scientific rationale for this choice of model structure and identify points where this choice could influence conclusions drawn. Please also describe the strength of the scientific basis underlying the key model assumptions.

Line 117 – should ‘MD’ be ‘IMD’ at the end of this line?

Please also see reviewer comments (below) regarding additional details required in respect of your methodology, which we agree with.

TABLES

Throughout please ensure that all tables are affiliated to an appropriate title and caption which clearly describes their content without the need to refer to the text.

Throughout, please ensure that all abbreviations including those used for statistical reporting are clearly defined for the reader.

Throughout, please clearly define the numerical values within the tables including those within brackets.

Throughout please indicate whether your analyses are adjusted (or not) and in the event that adjusted analyses are presented please detail the factors adjusted for (in an appropriate footnote/caption) and to help facilitate transparent data reporting, please also present the unadjusted analyses for comparison.

Throughout tables could be presented a little more accessibly and informatively for the reader. For example, when referring to means there are no units of measurement but some refer to weight, others to total number. This could be included in the columns headers for example, ‘mean sugar (g)’ presumably? 

Suggest separating upper and lower CI bounds with commas to conserve space and prevent data being split across rows. 

PLOS Medicine requires that where 95% CIs are reported, p values are also reported, please include and report as >0/001 and where higher the exact p value as 0.002, for example. If not reporting p values, for the purpose of transparent data reporting, please clearly state the reasons why not.

FIGURES

Throughout please cite figures (and tables) as outlined here https://journals.plos.org/plosmedicine/s/figures#loc-how-to-submit-figures-and-captions

Throughout (including the supporting information) please consider avoiding the sue of green and/or red so as to make your figures more accessible to those with color blindness.

Please ensure that all figures are affiliated to an appropriate title and caption that clearly describes the figure content without the need to refer to the text. Please ensure that any abbreviations including those used for statistical reporting are clearly defined for the reader.

Figure 2 – please include a caption. Please clearly state the meaning of the different colour shading. Please clearly define the meaning of the dots and the lines for the reader. 

DISCUSSION

Please remove all sub-headings form the discussion such that it reads as continuous prose and please ensure that you organize the Discussion as follows: a short, clear summary of the article's findings; what the study adds to existing research and where and why the results may differ from previous research; strengths and limitations of the study; implications and next steps for research, clinical practice, and/or public policy; one-paragraph conclusion.

Please remove the financial disclosure and competing interests statement from the end of the discussion and include only in the manuscript submission form when you re-submit your manuscript. In the event of publication, this information will be compiled as metadata.

REFERENCES

For in text reference callouts please remove the spaces between different citations, for example line 67 should read ‘[3,4]’.

In the bibliography please ensure that you list up to but no more than 6 author names followed by et al.

For all web references please ensure you include an, ‘Accessed [date].’

Journal name abbreviations should be those listed in the National Center for Biotechnology Information (NCBI) databases.

SUPPORTING INFORMATION

Please ensure that you provide titles and legends for each individual table and figure in the Supporting Information which follow the guidance detailed above for the main manuscript. 

In the published article, supporting information files are accessed only through a hyperlink attached to the captions. For this reason, you must list captions at the end of your manuscript file. You may include a caption within the supporting information file itself, as long as that caption is also provided in the manuscript file. Do not submit a separate caption file.

Please cite your Supporting Information as outlined here: https://journals.plos.org/plosmedicine/s/supporting-information

For in-text reference callouts please place citations in square parentheses separate by commas. For example, [1,3,6] or [1-3]. Please check and amend throughout all sub-sections of the manuscript and supporting files.

Please ensure that other referencing follows the format as detailed above for the main manuscript.

As for the main manuscript where reporting 95% CIs please also report p values as detailed above, if not please clearly state the reasons why not.

COMMENTS FROM THE REVIEWERS:

Reviewer #1: The paper models the potential on BMI and dental caries that occurred due to small reductions in sugar from beverages in UK households following a levy. The initial findings are promising that the intervention will effectively achieve increases reductions in both BMI and dental carries. While there are limitations to the methodology in that real dietary intake was not measured, the authors have described these in the limitations section. The analyses does not measure the impact on purchases not brought into the home, as sugary drinks are often consumed outside of the home, the authors may with to comment on this, particularly in relation to teenagers given the high intake of sugar in this age group.

Reviewer #2: Overview

This paper uses interrupted time series analyses and lifetable modelling to evaluate the effects of the UK soft drink industry levy on changes in sugar consumption, BMI, dental caries and QALYs in children and adolescents. This is an important topic, and the authors provide strong justification for their study. The paper is interesting, well written and thorough. I have made some minor comments and suggestions.

Comments

Introduction: Says that drinking with <0.5% ABV are exempt, I think this should be >1.2% ABV.

Results: I might have missed them, but I cannot see the full time series model specification and results in the main document or supplementary materials. It would be helpful to include these.

Discussion: This is excellent, and clearly explains every key limitation of the study that I noted from the methods.

Typo: MD line 117.

Note for editor: Thank you for the opportunity to review this paper. I would just like to note that I was unable to review the lifetable model in detail as I am not familiar with these methods. To the best of my knowledge, the methods appear robust.

Reviewer #3: Thank you to the authors for submitting this ambitious paper on a clearly very important topic - I enjoyed reading it. Although very well written, my overall opinion is that the paper is trying to do too much. As a consequence of this, several parts of the paper require far more explanation and detail. I also have concerns about some of the methods used and the similarity to another paper by most of the same authors. I feel as though the paper does have the potential for publication in this journal, however in my opinion a number of relatively substantive edits are needed. 

My comments are organised in the order in which they appear in the manuscript, and I have noted whether they are 'major' or 'minor'.

Line 62: Reference to the Rogers et al unpublished paper. Do the authors expect this paper to be published imminently? (Minor)

Lines 81 - 83: This sentence explaining the results from the 2016 systematic review is a bit confusing. It implies that the studies included in the review had one of two (quite distinct) outcomes. I would like another sentence or two explaining the mechanisms behind these findings from the authors (Minor)

Lines 100 - 103: From reading this, it seems as though the only difference between ITS analysis in the Rogers et al unpublished study and this present study is the fact that you stratified by IMD and used Great Britain rather than England. These analyses are very similar, and to me could have quite easily been sensitivity analysis in the Roger et al unpublished study. I'm not sure these two very similar analyses can be justified as two separate papers (Major)

Lines 114 - 115: I'm not 100% convinced by the choice of control - has this been used in any other studies aside from the Rogers et al unpublished study? Are there any other alternative choices of control that could have been used? (Minor)

Lines 117 - 188: The Kantar dataset needs to be explained in a lot more detail (Major)

Lines 117 - 188: The levels of missing data need to be stated in order to get an idea of the extent of the issue (Major)

Line 130: I'm a bit confused by the weights. Could you please elaborate on exactly how they were used? 

Line 139: Maybe it's my memory failing me, but I don't recall any widespread changes to consumer behaviour in December 2019 due to Britain's exit from the EU. It's not a big issue because COVID came soon after and this is clear justification for your cut-off point, but it seems a bit odd to me (Minor)

Line 147: I understand your use of the AIC to choose the optimal model, but I would like a bit more explanation regarding the number and type of different models you used when selecting your model.

Lines 157 - 166: I'm really not convinced by the life table modelling approach. Obesity (especially child/adolescent obesity) is a particularly difficult clinical area to model, with lifetime state transition models commonly used. I think there needs to be far more justification about why this highly simplified model structure is appropraite for this complex issue (Major)

Figure 1: I'm confused about how exactly you measured the impact of both BMI and Dental Caries on Quality of Life. Did you about for the fact that there may be an interactive effect? (Minor)

Lines 202 - 203: I appreciate the use of the EQ-5D to measure utility, however I think the authors need to state up front that these utility weights are based on a sample of adults rather than children. Although the EQ-5D-Y has been developed, there are no current population norms for either the 3L or 5L version, let alone utility scores by age, sex and IMD quintile (Major)

Line 203: I think it could be worth noting that the health state utilities gathered from reference number 49 and 50 are pretty out of date (Minor)

Lines 214-215: Need far more explanation about how exactly the life table model was implemented (Major)

Line 219-220: Can the authors comment on the plausibility of the changes in sugar purchased being maintained over the life course? Is there any evidence to back up this assumption? (Minor)

Line 226 - 228: This linear regression model needs far more explanation. What was the exact model specification? What was the exact sample it was run on? (Major)

Figure 2: I this figure difficult to interpret - it needs more explanation.

Line 309: The headline figure of "19,500 QALYs" is certainly eye catching, but I think it needs to be put into context (in relation to the child/adolescent population in England/UK). I'm not sure whether this is a small or large relative effect! (Major)

Lines 320-321: As before, I think more justification is needed for two separate papers (Minor)

Line 377: I think you could also discuss the representativeness of the dataset to the English/UK population - how big is the internal vs external validity trade off in this case? (Minor)

Reviewer #4: The study analyzes changes sugar purchased associated with the UK soft drink industry levy and modeled the effect on health in children and adolescents. The study is very relevant but needs to be revised.

I first recommend to change "effects or impact" by changes in or associations as in the absence of an experimental design, causal effects cannot be claimed for both the empirical estimation on sugar purchased and the model to estimate changes in dental caries and body mass index.

I understand that toiletries can be an adequate control but the authors should provide a convincing justification. It is crucial to test if the magnitude and trend prior to the implementation are not significantly different compared to SSB. If the test fails, toiletries is not an adequate control group. This can be done with ITSA models with a control group.

If the data set is at the household level, how was the specification for ITSA? Was it adapted to an ITSA? It is unclear as it seems that they are treating the data as time series given the autocorrelation test. I suggest to include the specification and provide a reference for ITSA models. 

Why are they modeling only sugar purchases and not also volume purchased? Did all brands reformulated?

Is Kantar data nationally representative? What do you mean by "proprietary weight?

Why using an area level variable for socioeconomic status (SES) instead of household SES? How many missing data were in the data set? More than 10%? I suggest to provide results with and without the missing data. 

The data is at the household level, I suggest to model purchases of sugar by adult equivalent or per capita and to adjust for household composition. Also, there are time varying macroeconomic variables that are associated with purchases, this should be included also.

In the results section, table 1, it is unclear how the estimations were derived from the ITSA model and if they included changes in level and slope.

Results for the third quintile are positive and unexpected. This should be addressed.

The time period covers the COVID-19 pandemic, if as in many countries there were changes in consumption patterns, this should be discussed and included in the model.

For the modeling exercise, what is the assumption for the duration of the intervention?

[LINK]

---

## [Decision Letter · Decision Letter 2]

12 Jan 2024

Dear Dr. Scarborough,

Thank you very much for submitting your manuscript "Impact of the UK Soft Drinks Industry Levy on health and health inequalities in children and adolescents in England: an interrupted time series analysis and population health modelling study" (PMEDICINE-D-23-02980R2) for consideration at PLOS Medicine. 

Your paper was evaluated by all the editors here. It was also discussed with an academic editor with relevant expertise, and sent to the statistical reviewer for re-review. The comments are appended at the bottom of this email and any accompanying reviewer attachments can be seen via the link below:

[LINK]

In light of the comments, we have invited you to undertake a further major revision. We cannot make any decision about publication until we have seen the revised manuscript and your response, and we plan to seek further re-review by the statistical reviewer. 

We expect to receive your revised manuscript by Feb 02 2024 11:59PM. Please email us (plosmedicine@plos.org) if you have any questions or concerns.

We look forward to receiving your revised manuscript. 

Sincerely,

Philippa Dodd, MBBS MRCP PhD

PLOS Medicine

plosmedicine.org

pdodd@plos.org

COMMENTS FROM THE ACADEMIC EDITOR

I think now it is much clearer, their methods, their controls, etc. I think they answered carefully all the comments. I liked the new appendix and the clarity of their analysis. However, I concur that a major revision is needed to address the statistical reviewer comments.

COMMENTS FROM THE EDITOR-IN-CHIEF

The authors must discuss the strength of the causal inference and provide details of how secular confounding was ruled out.

COMMENTS FROM THE EDITORS

GENERAL

Thank you for your detailed responses to previous editor and reviewer comments. Please see below for further comments including from the statistical reviewer which we require that you address in full. 

Due to the concerns raised by the statistical reviewer and the potential impact on the overall message conveyed by your paper, we have invited a further major revision for you to address these.

When re-submitting your manuscript, please include a clean, untracked and unhighlighted version with all changes accepted as well as a tracked version.

DATA AVAILABILITY STATEMENT

Thank you for updating your statement. Please also include a URL for Kantar and a contact email address for data enquiries.

AUTHOR SUMMARY

Thank you for including an author summary.

As detailed previously, the author summary should constitute a non-technical summary of your research which makes findings accessible to a wide audience including both scientists and non-scientists. The author summary should consist of 2-3 succinct bullet points under each of the headings. The text should be distinct from the abstract. 

In the final bullet point of ‘What Do These Findings Mean?’, please describe the main limitations of the study in non-technical language.

Please revise for brevity and improved accessibility to the non-scientist – we suggest removing all statistical information and describing your data in simple language.

METHODS and RESULTS

We agree with the statistical reviewer (please see below) that further explanation of the life table model should be included in the main text (as opposed to the supplementary files). Please include.

Lines 278, 287 & 300 – detail an ‘error’ related to citation of your supplementary files, perhaps confusion within the citation manager? Please check carefully throughout and amend this list is not exhaustive.

FIGURES 

Figure 1 – please ensure that all abbreviations and symbols (including those for units of measurement) are clearly defined in a footnote.

ACKNOWLEDGEMENTS

Please remove this statement in light of there being ‘none’.

SUPPORTING INFORMATION

Throughout, please cite and label your Supporting Information (including individual tables and figures within the documents) as outlined here: https://journals.plos.org/plosmedicine/s/supporting-information

S2 text – please ensure that all abbreviations used in tables and figures are clearly defined in a footnote or the caption.

S2 text, S5 Fig – please clearly state the meaning of the lines over the bars for the reader.

S3 text - please amend formatting to figure labels S1 Fig as opposed to Figure S1. 

COMMENTS FROM THE STATISTICAL REVIEWER:

Reviewer #3: Thank you to the authors for replying in detail to the comments from myself and the other reviewers. There are still several (fairly major) aspects of the paper that I think need to be addressed before publication can be considered. 

1. Similarity of this paper to the Rogers et al paper

I am still not convinced that this paper is different enough in its current format from the Rogers et al paper to justify a standalone paper. I will leave this issue to the journal editor(s).

2. Missing Data

Given the responses to my comment regarding missing data and the comment from Reviewer 4, I think a complete should definitely be presented as part of a sensitivity analysis. The fact that the missing data is by coincidence very much restricted to the pre-announcement phase of the study and therefore has a significant effect on the counterfactual is not a good enough excuse for not including this sensitivity analysis. 

3. How the optimal ITSA model was chosen

I am still not clear how the optimal ITSA was chosen on the basis of the AIC. Please be explicit and state how many different models were run, and the autoregressive order (p), moving average order (q) and AIC of the final model. Include this information either in the main text or (more likely) in the supplementary materials. 

4. Lifetable Modelling Approach

I am still not convinced by the life table modelling approach used in the paper given the complexity of obesity as a disease area. Please include further discussion of the limitations of this life table modelling approach in the context of obesity.

5. Combining QALY impact of BMI and Dental Caries in a "multiplicative way"

The measurement and valuation of utility values is one of my research interests, and I've personally never heard of utilities being combined in a "multiplicative way". Please further justify this method, including references to both theoretical and applied paper (if applicable).

6. Explanation of the life table model in the main text

I still think further explanation of the life table model is needed in the main text, not the supplementary file. This is a key part of the paper.

[LINK]

---

## [Decision Letter · Decision Letter 3]

28 Feb 2024

Dear Dr. Scarborough,

Thank you very much for re-submitting your manuscript "Impact of the UK Soft Drinks Industry Levy on health and health inequalities in children and adolescents in England: an interrupted time series analysis and population health modelling study" (PMEDICINE-D-23-02980R3) for review by PLOS Medicine.

I have discussed the paper with my colleagues and the academic editor and it was also seen again by the statistical reviewer. I am pleased to say that provided the remaining editorial and production issues are dealt with we are planning to accept the paper for publication in the journal.

[LINK]

If you have any questions in the meantime, please contact me at pdodd@plos.org or the journal staff on plosmedicine@plos.org.  

We look forward to receiving the revised manuscript by Mar 06 2024 11:59PM.   

Kind regards,

Pippa

Philippa Dodd, MBBS MRCP PhD

PLOS Medicine

plosmedicine.org

pdodd@plos.org

Requests from Editors:

GENERAL

Thank you for your detailed and considered responses to previous editor and reviewer comments. Please see below for further comments which we require you address prior to publication.

DATA AVAILABILITY

PLOS is committed to Open Science practices and transparent data reporting. In line with those commitments, PLOS Medicine has strict policies regarding data availability.

We require that the company name (Kantar) and the URL are provided. It is vital that the reader is made aware of the origins of the dataset used for your study. We also require a contact email address for data inquiries, please note that this cannot be a study author. Please update your statement in the manuscript submission form.

* This is a prerequisite to publication, and we cannot proceed without this information *

Please also include, as supporting information, the relevant documentation from Kantar which states their previous and now updated policy. 

AUTHOR SUMMARY

Line 70 – sentence beginning, ‘while children and…’ suggest making into a separate bullet point beginning, ‘Children and…’ as we think is an important point to make and should be emphasized.

METHODS and RESULTS

We agree with the statistical reviewer, please see below, that it would be beneficial to include the results without multiple imputation as supporting information, alongside a brief note in the main text of the results section.

Line 289 – please correct the ‘table 1 error. Reference source not found’.

DISCUSSION

As per the statistical reviewer’s request, please discuss the limitations of the lifetable model.

SUPPORTING INFORMATION

As above, please include Kantar’s data policies.

As above, please include results without imputation as supporting information, per the statistical reviewer’s request.

S2 Text:

1) Page 5 – please define ‘DMFT’ at first use here, apologies if I have missed it previously.

2) Figures – please define ‘IMD’ in the footnotes of all figures.

SOCIAL MEDIA

To help us extend the reach of your research, please detail any X (formerly Twitter) handles you wish to be included when we tweet this paper (including your own, your coauthors’, your institution, funder, or lab) in the manuscript submission form when you re-submit the manuscript.

Comments from Reviewers:

Reviewer #3: Thank you to the authors for replying to my concerns. I have a couple of (very) minor things that I think should be changed before publication:

Missing Data

Thank you to the authors for providing these additional results. Please can the additional results provided by the authors (without multiple imputation) be included as part of the supplementary materials and a sentence added to main text noting that these results are included. I think it's important that these are included and referenced in the paper. 

Lifetable Modelling Approach 

As noted in my previous review (R3), please can the authors include a sentence or two discussing the potential limitations with their modelling approach in the main text.

After these minor changes have been done I'll be happy to approve the manuscript for publication.

[LINK]

---

## [Editor Report · Decision Letter 4]

6 Mar 2024

Dear Dr Scarborough, 

On behalf of my colleagues and the Academic Editor, Professor Barry Popkin, I am pleased to inform you that we have agreed to publish your manuscript "Impact of the UK Soft Drinks Industry Levy on health and health inequalities in children and adolescents in England: an interrupted time series analysis and population health modelling study" (PMEDICINE-D-23-02980R4) in PLOS Medicine.

We thank you for your responsiveness to previous comments. Before your manuscript can be formally accepted you will need to complete some formatting changes, which you will receive in a follow up email. Please be aware that it may take several days for you to receive this email; during this time no action is required by you. Once you have received these formatting requests, please note that your manuscript will not be scheduled for publication until you have made the required changes.

PRESS

Kind regards,

Pippa 

Philippa C. Dodd, MBBS MRCP PhD 

PLOS Medicine

pdodd@plos.org